# Electrically induced and detected Néel vector reversal in a collinear antiferromagnet

J. Godinho [1,2], H. Reichlová[1,3], D. Kriegner [1,4], V. Novák[1], K. Olejník[1], Z. Kašpar[1], Z. Šobáň[1], P. Wadley[5], R.P. Campion[5], R.M. Otxoa[6,7], P.E. Roy[6], J. Železný[1], T. Jungwirth[1,5] & J. Wunderlich[1,6]

Antiferromagnets are enriching spintronics research by many favorable properties that include insensitivity to magnetic fields, neuromorphic memory characteristics, and ultra-fast spin dynamics. Designing memory devices with electrical writing and reading is one of the central topics of antiferromagnetic spintronics. So far, such a combined functionality has been demonstrated via 90° reorientations of the Néel vector generated by the current-induced spin orbit torque and sensed by the linear-response anisotropic magnetoresistance. Here we show that in the same antiferromagnetic CuMnAs films as used in these earlier experiments we can also control 180° Néel vector reversals by switching the polarity of the writing current. Moreover, the two stable states with opposite Néel vector orientations in this collinear antiferromagnet can be electrically distinguished by measuring a second-order magnetoresistance effect. We discuss the general magnetic point group symmetries allowing for this electrical readout effect and its specific microscopic origin in CuMnAs.

[1] Institute of Physics, Czech Academy of Sciences, Cukrovarnická 10, 160 00 Prague 6, Czech Republic. [2] Faculty of Mathematics and Physics, Charles University in Prague, Ke Karlovu 3, 12116 Prague 2, Czech Republic. [3] Institut für Festkörper- und Materialphysik, Technische Universität Dresden, 01062 Dresden, Germany. [4] Max Planck Institute for Chemical Physics of Solids, 01187 Dresden, Germany. [5] School of Physics and Astronomy, University Of Nottingham, NG7 2RD Nottingham, United Kingdom. [6] Hitachi Cambridge Laboratory, Hitachi Europe LTD, JJ Thomson Avenue, Cambridge CB3 0HE, United Kingdom. [7] Donostia International Physics Center, Paseo Manuel de Lardizabal 4, Donostia-San Sebastian 20018, Spain. Correspondence and requests for materials should be addressed to J.G. (email: godinho@fzu.cz) or to J.W. (email: jw526@cam.ac.uk)

**E**lectrical detection of the 180° spin reversal, which is the basis of the operation of ferromagnetic memories[1], has been among the outstanding challenges in the research of anti-ferromagnetic spintronics[2–5]. Analogous effects to the ferromagnetic giant or tunneling magnetoresistance have not yet been realized in antiferromagnetic multilayers[6]. Anomalous Hall effect (AHE), which has been recently employed for spin reversal detection in non-collinear antiferromagnets, is limited to materials that crystalize in ferromagnetic symmetry groups[6–11]. Here we demonstrate electrical detection of the 180° Néel vector reversal in CuMnAs, which comprises two collinear spin sub-lattices and belongs to an antiferromagnetic symmetry group with no net magnetic moment. We detect the spin reversal by measuring a second-order magnetotransport coefficient whose presence is allowed in systems with broken space inversion symmetry. The phenomenology of the non-linear transport effect we observe in CuMnAs is consistent with a microscopic scenario combining anisotropic magnetoresistance (AMR) with a transient tilt of the Néel vector due to a current-induced, staggered spin-orbit field[6,12,13]. We use the same staggered spin-orbit field, but of a higher amplitude, for the electrical switching between reversed antiferromagnetic states, which are stable and show no sign of decay over 25 h probing times.

Before presenting the experimental data, we first elaborate in more detail on a microscopic mechanism that gives the seemingly counter-intuitive possibility for detecting 180° spin reversal in a collinear antiferromagnet comprising two chemically identical spin sublattices. The mechanism is illustrated in Fig. 1a–c. It is based on the observation that the sites occupied by nearest-neighbor Mn atoms in CuMnAs are locally non-centrosymmetric inversion partners. This implies that electrical current induces a non-equilibrium spin polarization with opposite sign on the two sites[12,13]. Simultaneously, the inversion-partner Mn sites belong to opposite spin sublattices of the bipartite Néel order ground state[12,13]. As the staggered current-induced polarization, and corresponding staggered effective field, are commensurate with the Néel order, the antiferromagnetic moments can be deflected by relatively weak currents. The electrically induced Néel vector deflection combined with AMR can then yield a second-order magnetotransport effect applicable for detecting the 180° Néel vector reversal. Later in the discussion part, we show that this microscopic mechanism is consistent with a general symmetry-based picture in which the spin reversal detection by a second-order

magnetoresistance is allowed in antiferromagnets ordering in magnetic point groups with broken time and space inversion symmetry. In the next paragraph, we continue by illustrating the experimental implementation of this detection technique.

We recall that in the tetragonal lattice of CuMnAs, the staggered field generated by a current applied in the $a − b$ plane is along the in-plane axis oriented perpendicular to the current, as highlighted in Fig. 1a[13]. Considering this geometry, we sketch in Fig. 1b a set-up for detecting the 180° reversal of the Néel vector pointing 45° rotated to the $x$ axis of the current. Here the reversal is measured by the longitudinal current-dependent resistance $\delta R_{xx}$. Another example of the measurement set-up is shown in Fig. 1c, where we sketch the detection of the reversal of the Néel vector pointing along the $x$ axis via the current-dependent transverse resistance $\delta R_{xy}$. (For more details on the detection scheme, see Supplementary Note 2 and Supplementary Figure 2).

To perform the experiment we need, apart from the readout method, also a tool allowing us to reverse the Néel vector in CuMnAs. For this, we employ again the current-induced staggered spin-orbit field. Unlike the weaker currents applied to induce transient changes of the Néel vector angle during readout, for writing we apply higher amplitude currents and the bistable 180° reversal is controlled by flipping the polarity of the writing current[3,14]. We note that the analogous writing method was used in earlier studies of 90° Néel vector reorientation in CuMnAs and Mn₂Au, controlled in this geometry by two orthogonal writing current lines and detected by the linear-response AMR[12,13,15–18].

In this article, we show current-pulse polarity-dependent 180° switching between two, energetically equal antiferromagnetic states with opposite Néel vector orientations. The reversed states are electrically distinguished by measuring a second-order magnetoresistance effect.

## Results

**Experimental structures and measurement technique.** Devices used in our experiments were fabricated from a 10 nm thick CuMnAs film grown by molecular beam epitaxy on a GaAs substrate[19] and protected by a 3 nm Pt layer. The sheet resistance of the stack is 100 Ω. Note that the Pt cap provides additional Joule heating when the writing pulses are applied to the stack. The Joule heating assists but is not governing the deterministic, polarity-dependent switching. Further discussion of the structure

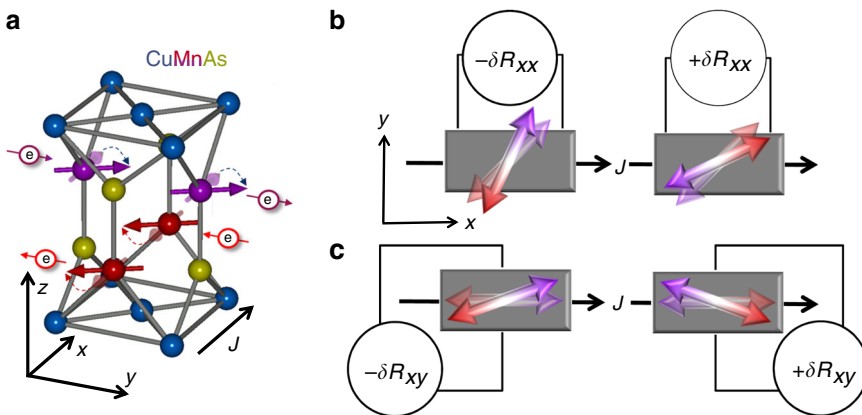

**Fig. 1** Microscopic mechanism of the second-order magnetoresistance. **a** Probing current (black arrow) generates staggered non-equilibrium spin polarization (red and purple electron symbols with arrows) that causes transient deflection of the antiferromagnetic moments (thick red and purple arrows on Mn sites). **b** The 180° reversal of the Néel order probed by the current-dependent resistance $\delta R_{xx}$, associated with the electrically induced deflection of antiferromagnetic moments (double-arrows) combined with AMR, for equilibrium antiferromagnetic moments (semi-transparent double-arrows) aligned at an angle 45° from $x$ axis of the probing current. **c** Same as **b** for $\delta R_{xy}$ and equilibrium antiferromagnetic moments aligned with $x$ axis

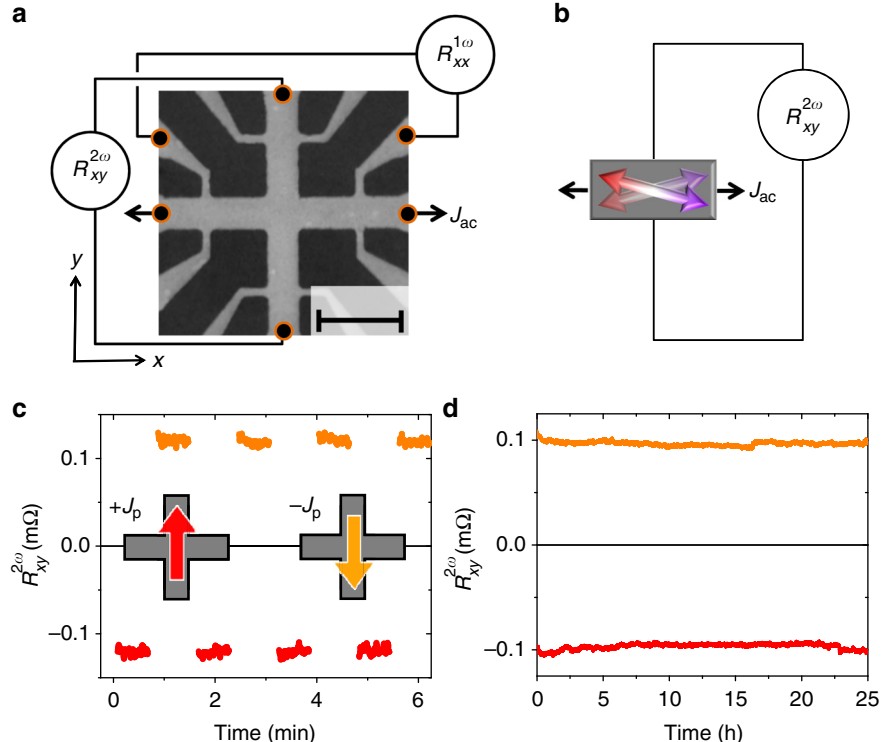

**Fig. 2** Electrical detection of 180° reversal of the Néel order in CuMnAs. **a** Scanning electron micrograph of the cross-bar device with contacts allowing to measure longitudinal and transverse resistances along $x$ and $y$ axes. The added schematics correspond to the measurement set-up for the simultaneous detection of the transverse second-harmonics signal $R_{xy}^{2\omega}$ and the longitudinal first-harmonics signal $R_{xx}^{1\omega}$. The scale bar length corresponds to 10 μm. **b** Schematics of measurement set-up with an alternating probing current $J_{ac}$ along $x$ axis and second-harmonic voltage detected along $y$ axis, giving $R_{xy}^{2\omega}$. Double-arrows illustrate a microscopic mechanism in which $J_{ac}$ generates alternating deflections of the antiferromagnetic moments. **c** 20 ms long pulses of the writing current $J_p = 11$ mA ($j_p \sim 10^7$ A cm$^{-2}$ in CuMnAs) along the ±$y$ direction (red/yellow arrows) are applied to set the Néel vector along the ±$x$ axis. Second-harmonic transverse resistance $R_{xy}^{2\omega}$ is measured with a probing current $J_{ac} = 2$ mA ($j_{ac} \sim 10^6$ A cm$^{-2}$) applied along the $x$ axis. **d** Same as **c** for one writing pulse along +$y$ axis and one subsequent pulse along −$y$ axis and 25 h measurement of the stability of the second-harmonic probing signal

of our materials and measurements on a CuMnAs film capped with AlO$_x$ are presented in the Supplementary Notes 1, 2 and 3.

The wafers were patterned into Hall cross structures with added contacts to enable simultaneous detection of transverse and longitudinal signals, as shown on the scanning electron micrograph of the device in Fig. 2a. The longitudinal (linear-response) resistance of the structure is approximately 1 kΩ. In our detection experiments, a low frequency ($\omega/2\pi = 143$ Hz) probing current $J_0\sin(\omega t)$ with an effective value of $J_{ac} = J_0/\sqrt{2}$ is applied to our device. We use lock-in amplifiers to measure simultaneously first-harmonic ($1\omega$) and second-harmonic ($2\omega$) components of the voltage signals. The former detects the linear-response AMR. The latter probes the second-order magnetotransport response, which we associate, following the mechanism in Fig. 1, with AMR combined with a periodic variation of the current-induced staggered field and the corresponding periodic Néel-vector deflection (see Fig. 2b). Note that the second-order transport effects would also appear, in principle, in the zeroth harmonic voltage component. In our off-resonance experiments, however, this component is difficult to extract from the measurement noise. The second-harmonic component, on the other hand, can be accurately measured by employing the homodyne detection method. For more details, on our experimental methods see Supplementary Note 2.

**Electrical generation and detection of the 180° reversal.** Key results of our experiments are summarized in Fig. 2c, d where the

plotted second-harmonic resistance is obtained by dividing the corresponding second-harmonic voltage by the probing current $J_{ac}$. In Fig. 2c, we first sent a 20 ms long writing pulse $J_p$ of amplitude 11 mA (corresponding to a current density $j_p \sim 10^7$ A cm$^{-2}$ flowing through the CuMnAs film) along the $y$ direction to set the Néel vector along the $x$ axis. We then measure for 40 s the resulting second-harmonic transverse resistance $R_{xy}^{2\omega}$ with a probing current $J_{ac} = 2$ mA applied along the $x$ axis. Next, we flip the polarity of the writing pulse in order to reverse the Néel vector and again measure $R_{xy}^{2\omega}$ with the same probing current. The sequence is repeated several times. As expected for the second-order magnetoresistance mechanism described in Fig. 1, we observe reproducible $R_{xy}^{2\omega}$ signals that are distinct for the two reversed states of the antiferromagnet. Figure 2d shows the same type of experiments for one of the reversal sequences but with the probing performed for each state over 25 h. The results highlight the stability of the detected 180° reversal signal, which exhibits no sign of decay at these long probing times.

The mechanism described in Fig. 1 suggests that we should not detect any reversal signal in $R_{xy}^{2\omega}$ if both the probing and setting currents are applied along the same direction ($x$ axis). This is because we set the Néel vector in this case collinear to the direction ($y$ axis) of the staggered effective field induced by the probing current and, therefore, no transverse deflection of the Néel vector is induced by the probing current. The picture is confirmed by the measured data shown in Fig. 3a where we apply a sequence of writing pulses along ±$y$ and ±$x$ directions, which are indicated by red/orange and dark/light green arrows in the

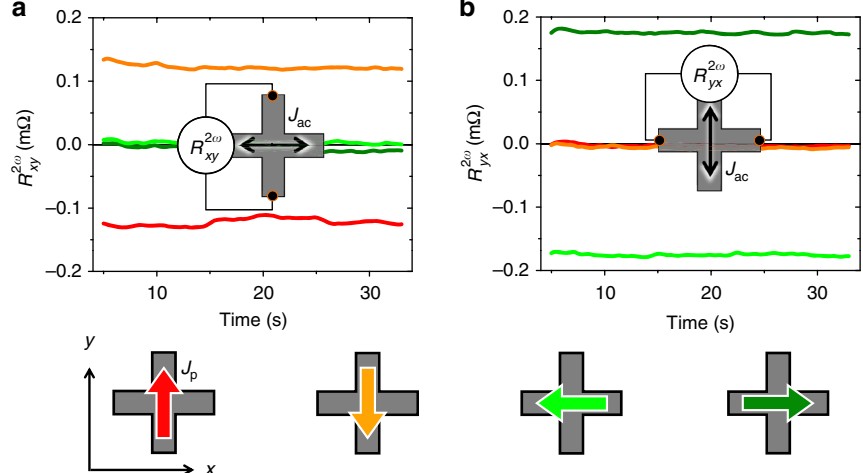

**Fig. 3** Symmetry of the second-harmonic signals. **a** $R_{xy}^{2\omega}$ readout for a sequence of writing pulses along $+y$, $+x$, $-y$, $-x$ directions. **b** Same as **a**, with $R_{yx}^{2\omega}$ readout (probing current along $y$ axis). Readout measurements in **a**, **b** start 5 s after the writing pulse

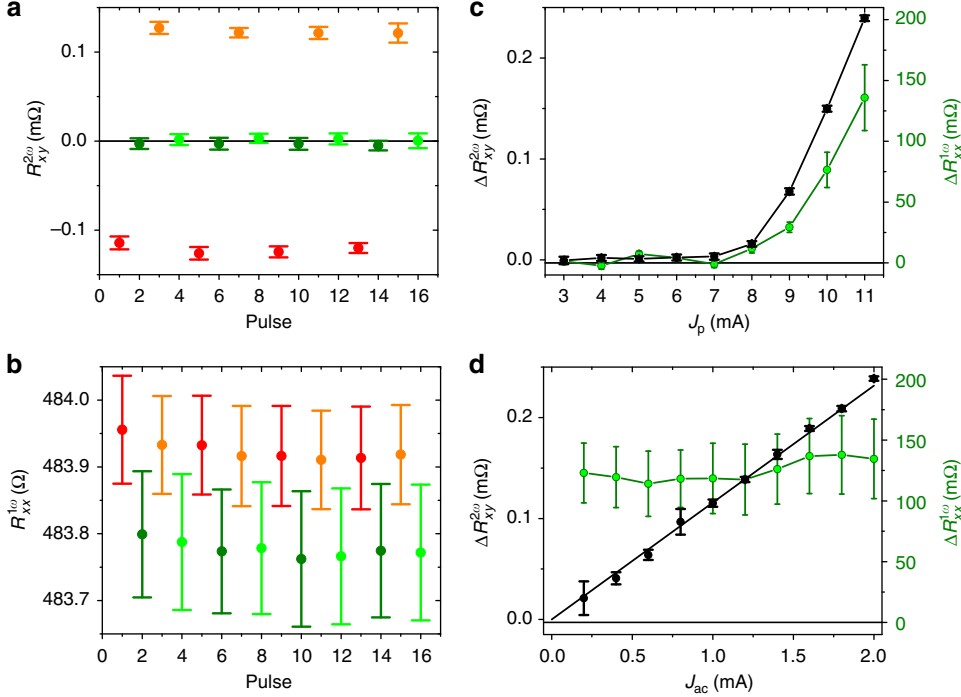

**Fig. 4** Comparison of second- and first-harmonic signals. **a** Second-harmonic $R_{xy}^{2\omega}$ signal measured for four sequences of writing pulses along $+y$, $+x$, $-y$, $-x$ directions. **b** Same writing sequences as in **a** probed with the first-harmonic $R_{xx}^{1\omega}$. **c** First- and second-harmonic signals measured as a function of the amplitude of the writing current pulse. **d** Dependencies of the first- and second-harmonic signals on the probing current $J_{ac}$. Probing signals are averaged over 30 s detection time starting 5 s after the writing pulse and error bars in **a**–**d** correspond to the standard deviation

device sketches in the figure. In Fig. 3a, the probing current is along the $x$ axis and $R_{xy}^{2\omega}$ can only detect the reversal between Néel vectors set along the $x$ axis by writing current pulses along the $y$ axis (red/orange). On the other hand, $R_{xy}^{2\omega}$ is negligible for states with Néel vectors set along the $y$ axis by writing current pulses along the $x$ axis (dark/light green). To highlight that it is indeed the second-order magnetoresistance probing that is not effective in this geometry and not an inability in our material to set the Néel vector along the $y$ axis, we rotate the detection set-up in Fig. 3b by 90°. When sending the probing current in the

$y$ direction and measuring $R_{yx}^{2\omega}$, we can now detect the reversal between the Néel vector states set along the $y$ axis (by writing current pulses along the $x$ axis). Consistently, the reversal of the antiferromagnetic order between states set along the $x$ axis (by writing current pulses along the $y$ axis) is not detectable by $R_{yx}^{2\omega}$, as also seen in Fig. 3b.

Since we can write four distinct states in our device with Néel vectors set along $\pm x$ and $\pm y$ axes, we can compare in Figs. 4a, b the second-harmonic signal with the first-harmonic AMR. We again show several pulsing sequences but, unlike Fig. 2c, we now

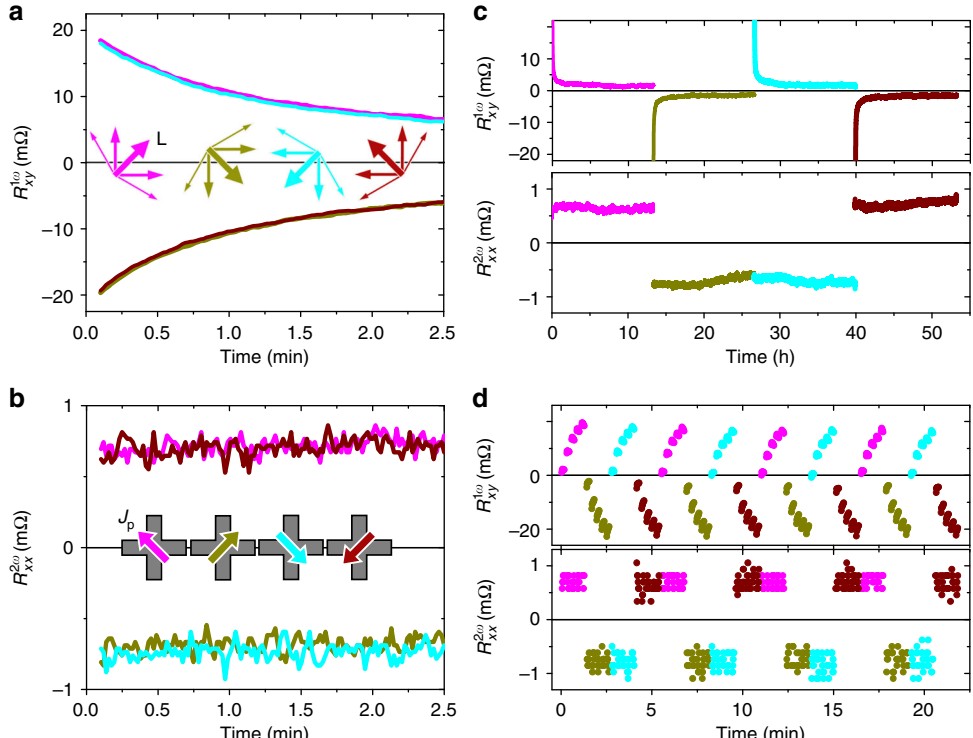

**Fig. 5** Time-dependence of first- and second-harmonic signals. **a, b** First-harmonic $R_{xy}^{1\omega}$ and second-harmonic $R_{xx}^{2\omega}$ detection for a sequence of writing current pulses along directions rotated by ±45° from the main cross axes with $j_p \sim 10^7$ A cm$^{-2}$ in the cross center. **c** Same as **a, b** for 12 h probing measurements after each writing pulse. **d** First- and second-harmonic signals measured when sending trains of five writing pulses along one direction before changing the pulsing angle. In all panels, probing starts 5 s after the writing pulse

rotate the pulsing current successively in steps of 90° within each sequence. The probing signals are averaged over 30 s detection time and error bars correspond to the standard deviation. Note that the larger error bars in the first-harmonic signal are typical for the longitudinal resistance in which the AMR generates only a small additional contribution (of <1% in the present experiment) on top of a large isotropic resistance of the device and where the latter can show, e.g., a significant drift with temperature[13]. Still we observe a clear switching signal in $R_{xx}^{1\omega}$, which, as expected for the linear-response AMR, allows us to distinguish states with Néel vectors set along the $x$ axis from states set along the $y$ axis, and gives no sensitivity to the 180° reversal. This, in turn, is detected in the same reorientation sequence by the second-harmonic signal (e.g., $R_{xy}^{2\omega}$ for Néel vector reversal along the $y$ axis). We also point out that the signs of the second- and first-harmonic signals in Fig. 4a, b are consistent with the microscopic picture of the second-order magnetoresistance originating from the combined effect of the current-induced deflection of the Néel vector due to the staggered spin-orbit field and the AMR.

**Comparison of second- and first-harmonic signals**. In Fig. 4c, we show the first- and second-harmonic signals as a function of the amplitude of the writing current pulses. Both signals show a common threshold of the writing current and a subsequent increase with increasing current amplitude. This implies that a similar amplitude of the staggered effective field and/or similar assisting Joule heating is required for setting any of the four measured Néel vector directions. In Fig. 4d, we show the dependencies of the first- and second-harmonic signals on the probing current. As expected for the linear-response transport coefficient, the first-harmonic resistance is independent of the

probing current, apart from a small scatter generated by the noisy $R_{xx}^{1\omega}$ signal. In contrast, the second-harmonic resistance increases with the probing current, consistent with the second-order nature of this magnetotransport coefficient. We estimate a minimum energy of ~1 μJ required for the second-harmonic readout in our experimental set-up, which was not optimized for minimizing the readout energy. However, increasing the probing current frequency would allow faster second-harmonic signal detection and correspondingly lower readout energies.

In Fig. 5, we show that the studied CuMnAs film shows an easy-plane-like behavior allowing us in principle to set the Néel vector in any in-plane direction. To illustrate this we apply the writing current pulses along directions rotated by ±45° from the main cross axes (as shown in the inset of Fig. 5b) by biasing both legs simultaneously[15]. The writing bias voltage is adjusted to generate again a current density $j_p \sim 10^7$ A cm$^{-2}$ in the cross center. Data in Fig. 5a, b are plotted for one sequence of cross-diagonal writing currents rotated successively in steps of 90°. For this writing geometry, the 90° Néel vector reorientation signal is detected in $R_{xy}^{1\omega}$, while the 180° reversal is probed by $R_{xx}^{2\omega}$.

The $R_{xy}^{1\omega}$ signal in Fig. 5a shows a significant decay over the probing time of 2.5 min starting 5 s after the writing pulse. This together with the increasing signal with the increasing writing current amplitude (Fig. 4c) points to a multi-domain nature of the active region of the device. The observation is consistent with results of previous 90° reorientation experiments utilizing both electrical probing and X-ray magnetic linear dichroism microscopy[13,15,20].

Remarkably, the counterpart 180° reversal signal in Fig. 5b, as well as the second-harmonic reversal signals in Figs. 2 and 3, show no decay from 5 s after the writing pulse when we initiate the electrical readout. We interpret this as follows: the

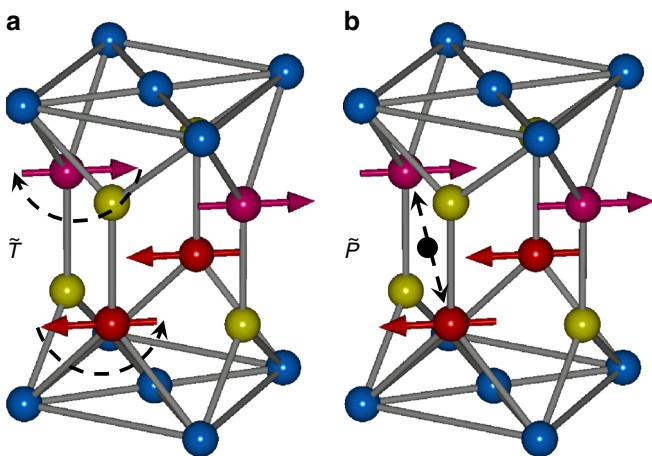

**Fig. 6** Time reversal and space inversion operation in the CuMnAs magnetic lattice. **a** The time reversal operation $\tilde{T}$ flips the magnetic moments, highlighting the broken time reversal symmetry. **b** Black point shows the inversion center of the space inversion symmetric non-magnetic lattice. However, the space inversion operation $\tilde{P}$ that exchanges magnetic atoms around the black point does not leave the antiferromagnetic lattice invariant, i.e., space inversion symmetry is broken in antiferromagnetic CuMnAs. The combined $\tilde{P}\tilde{T}$ operation remains a symmetry even in the antiferromagnetic state

first-harmonic signal measured 5 s after the pulse is already relatively low in the present experiment, corresponding to AMR of 0.08%. Note that in other CuMnAs films, microstructures, or setting conditions we can observe two orders of magnitude larger AMR signals. Magnetostriction is a mechanism that can explain the relaxation of the 90° reorientation signal in our thin film. Because of the locking of the antiferromagnet's lattice to the substrate, the system may tend to minimize its energy by breaking into domains with the Néel vector randomized within a semicircle around the initial setting direction. This would diminish the 90° reorientation signal toward zero.

On the other hand, the magnetostriction mechanism is even in the magnetic order parameter and, therefore, does not drive sign flips of the Néel vector. As a result, the randomization of the Néel vector is limited to the semicircle and, consequently, the 180° reversal signal would not drop below $(2/3)\pi$ times the signal corresponding to the single domain fully reversed state. From the comparison between Fig. 5a, b, we surmise that a significant randomization within the semicircle is already completed before we initiate the readout measurement ~10 s after the pulse was applied. The remaining small changes are then not observable within the experimental noise on top of the second-harmonic signal, which remains significant compared with its unrelaxed value. On the other hand, the changes are detectable in the remaining first-harmonic signal, which is strongly diminished compared with the first-harmonic signal of a fully switched state. As a result of the tendency of our antiferromagnetic structure to break into domains with Néel vector distributed within a semicircle around the initial setting direction, we observe the reproducible, stable easy-plane-like 180° reversals in the second-order magnetoresistance.

Further details on the comparison between the first- and second-harmonic signals are provided in Fig. 5c, d. First, we extended in Fig. 5c the probing time to 12 h to highlight the stability of the second-harmonic signal in comparison with the first-harmonic signal, which significantly decays in the present

structure. Consistently, we also see different characteristics of the first- and second-harmonic signals when sending trains of pulses along one direction before changing the pulsing angle (Fig. 5d). In the first-harmonic signal, we clearly resolve a memristive multi-level characteristics[13,15,20] because the small changes of the readout signal due to successive pulses within the train can be resolved on top of the overall weak (strongly relaxed) 90° reorientation signal. On the other hand, the small memristive effect of the successive pulses is not visible in the second-harmonic signal. As a result, the 180° reversal signals measured from 5 s after the setting pulse are stable and independent of history.

## Discussion
In the concluding paragraphs, we discuss the detection of the 180° reversal by the second-order magnetoresistance in antiferromagnets from a general symmetry perspective. Before turning to the non-linear magnetotransport detection, we first recall limitations of the linear-response effects in antiferromagnets. AHE corresponds to the linear-response magnetoresistance, $E_i = \rho_{ij}^{\text{odd}}(\mathbf{O})j_j$, which is odd under time reversal, i.e., $E_i = -\tilde{T}\rho_{ij}^{\text{odd}}(\mathbf{O})j_j = -\rho_{ij}^{\text{odd}}(-\mathbf{O})j_j$. Here $E$ is the electric field, $\tilde{T}\rho_{ij}^{\text{odd}}$ labels the resistivity tensor $\rho_{ij}$ transformed by the time-reversal operation $\tilde{T}$, $j$ is the current density, and $\mathbf{O}$ is the magnetic order parameter vector that breaks time reversal symmetry of the system. In antiferromagnets, AHE is allowed by symmetry only in a subset of the 122 magnetic point groups. These are the antiferromagnets that order in one of the 31 ferromagnetic symmetry point groups, i.e., can develop a net magnetic moment along some directions without changing the symmetry of the magnetic lattice[7]. Consistent with this symmetry argument, non-collinear weak-moment antiferromagnets $Mn_3Ir$, $Mn_3Sb$, or $Mn_3Ge$ have been recently identified to host the AHE[8–10]. Moreover, Liu et al.[11] have shown in $Mn_3Pt$ that piezoelectric strain can cause a magnetic phase transition from a non-collinear spin arrangement hosting the AHE to a collinear anti-ferromagnetic phase that does not belong to one of the 31 ferromagnetic symmetry point groups and, correspondingly, shows no AHE.

The antiferromagnetic lattice of CuMnAs has a broken time reversal symmetry in its magnetic point group, as illustrated in Fig. 6a. However, it is another example of an antiferromagnetic phase that does not belong to one of the ferromagnetic symmetry point groups. AHE is, therefore, excluded despite the broken time reversal symmetry. Namely, it is the combined space inversion—time reversal symmetry of CuMnAs (see Fig. 6a, b), which makes the AHE vanish in this antiferromagnet. We can see it from the above linear-response equation. Here the space inversion operation $\tilde{P}$ flips the sign of both electric field and current. This implies that applying space inversion and time reversal to the linear-response transport equation gives $\rho_{ij}^{\text{odd}} = -\tilde{P}\tilde{T}\rho_{ij}^{\text{odd}}$. On the other hand, the combined spatial inversion and time reversal symmetry of CuMnAs in conjunction with the Neumann's principle, linking the symmetries of a crystal to its physical properties, impose that $\rho_{ij}^{\text{odd}} = \tilde{P}\tilde{T}\rho_{ij}^{\text{odd}}$. The two conditions then yield $\rho_{ij}^{\text{odd}} \equiv 0$ by symmetry.

AMR is a complementary linear-response effect allowing to detect the direction of the order parameter in magnetic films. It has been detected in CuMnAs, as well as in FeRh, MnTe, or $Mn_2Au$ that all host a collinear fully compensated Néel order[13,16–18,21,22]. However, AMR corresponds to the linear-response magnetoresistance coefficient that is even under time reversal, $\rho_{ij}^{\text{even}}(\mathbf{O}) = \rho_{ij}^{\text{even}}(-\mathbf{O})$, i.e., gives the same electrical signal when reversing spins by 180°. This applies equally to any of

the 31 ferromagnetic symmetry point groups and also to any of the remaining 91 symmetry point groups of true antiferromagnets that do not allow for a net magnetic moment without changing the symmetry of the magnetic crystal.

By measuring the second-order magnetotransport coefficient, we can extend the detection of the 180° spin reversal from antiferromagnets within the 31 ferromagnetic point groups to the larger family of antiferromagnetic point groups with broken time reversal symmetry and no net moment allowed in the point group. In total, there is 59 of these broken time reversal symmetry antiferromagnetic point groups. However, there is an additional symmetry condition required for the presence of the second-order magnetotransport coefficient, which is the broken space inversion symmetry of the antiferromagnetic lattice. This can be seen by applying the space inversion operation $\tilde{P}$ on the second-order transport equation (odd under time reversal), $E_i = \xi_{ijk}^{\text{odd}} j_j j_k$, and recalling that $\tilde{P}$ flips sign of both the electric field and current. This implies for the second-order transport coefficient (odd under time reversal) that, $\xi_{ijk}^{\text{odd}} = -\tilde{P}\xi_{ijk}^{\text{odd}}$, which allows for a non-zero $\xi_{ijk}^{\text{odd}}$ only if the space inversion symmetry is broken.

As seen in Fig. 6, CuMnAs is one example from the 59 antiferromagnetic point groups with broken time reversal symmetry that has also broken space inversion symmetry in the magnetic crystal. In general, 48 out of the 59 antiferromagnetic point groups and 21 out of the 31 ferromagnetic point groups have broken space inversion symmetry, which makes the second-order detection method of the 180° spin reversal broadly applicable in antiferromagnets.

Symmetry arguments are the basis for analyzing whether a given effect can in principle exist in a certain class of materials. Its magnitude, on the other hand, is determined by the microscopic origin of the effect. Remarkably, the same combined space inversion—time reversal symmetry in CuMnAs, which excluded the AHE in this material, allows for the specific microscopic mechanism of the second-order magnetoresistance that combines current-induced deflection of the Néel vector with AMR. While our experiments are qualitatively compatible with this scenario, other microscopic mechanisms can contribute in CuMnAs or can govern the second-order magnetotransport detection of the 180° spin reversal in other antiferromagnets with broken time and space inversion symmetries.

## Methods

**Device fabrication and characterization**. Devices used for our experiments were fabricated from an epitaxial 10 nm thick tetragonal CuMnAs film grown by molecular beam epitaxy on a GaAs substrate[19] and covered in-situ by a Pt layer of a nominal thickness of 3 nm. Superconducting quantum interference device (SQUID) magnetometry measurements and X-ray diffraction (XRD) measurements on our material are shown in Supplementary figure 1. A reference film was grown simultaneously by masking part of the wafer during Pt evaporation. Additionally, a nominally 2.5 nm thick Al layer, which almost fully oxidizes when exposed to air, was deposited on top to protect the film against oxidation. Supplementary figure 3 shows the bipolar switching characteristics from a device patterned from the reference CuMnAs/AlOx film without the Pt layer.

Several devices of different sizes were prepared showing qualitatively the same results. Before patterning, the CuMnAs film was measured by SQUID magnetometry to exclude any ferromagnetic impurities, uncompensated moments, or proximity polarization in Pt. The data are shown in Supplementary Figure 1a.

XRD was employed to confirm the quality and thickness of the layers. Within the error bars, the measured CuMnAs thickness corresponds to the nominal value of 10 nm and the measured 3.6 nm thickness of Pt is also close to the nominal value. The measured Al cap thickness was found to be around 4 nm, i.e., slightly thicker than the nominal value of 2.5 nm of the deposited Al layer. We explain this by the oxidation of the Al cap. The measurements are shown in Supplementary Figure 1b.

Wafers were patterned into Hall cross devices, as shown in Fig. 2a, defined by electron beam lithography and patterned by argon plasma etch using a hydrogen silsesquioxane (HSQ) resist mask, which was removed afterwards. Electrical

contacts to the sample were defined by e-beam lithography, evaporation of Cr(5 nm)/Au(80 nm) bi-layer and followed by a lift-off process.

## Data availability

The relevant data are available within the article or from the authors on reasonable request.

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

## Acknowledgements

The authors acknowledge the EU FET Open RIA grant no. 766566, the Grant Agency of the Czech Republic grant no. 14-37427G, the Ministry of Education of the Czech Republic grant no. LM2015087 and LNSM-LNSpin, and the ERC Synergy grant no. 610115.

## Author contributions

J.G., T.J., H.R., D.K., and J.W. conceived and designed experiments. J.G., H.R., D.K., and Z.K. performed experiments. J.G., H.R., D.K., K.O., P.W., R.M.O, P.E.R., J.Ž., T.J., and J.W. analyzed data. V.N., R.P.C., J.G., H.R., and Z.Š. contributed materials and devices.

J.W., H.R., D.K., and T.J. wrote the paper. J.W. planned the study. All authors discussed the results.

## Additional information

**Competing interests:** The authors declare no competing interests.

