## [Peer Review File · Nature Communications]

Reviewers' comments:

Reviewer #1 (Remarks to the Author):

Manipulation of spins in antiferromagnets is the key task of the emerging antiferromagnetic (AFM) spintronics. However, it is rather more difficult compared with the steering of the ferromagnetic magnetization. Up to now, the exchange spring effect [Nat. Mater. 10, 347 (2011)], the Neel spin-orbit torque [Science 351, 587 (2016)] and the electric field [Nat. Electron. 1, 172 (2018)] have been demonstrated to effectively control the AFM order parameter.

The detection of spin reversal in antiferromagnets is even more challenging due to their tiny macroscopic moments. Fortunately, the Berry-curvature-induced anomalous Hall effect has enabled the successful detection of the spin reversal via the sign of the Hall resistance in non-collinear antiferromagnets including both hexagonal "soft" antiferromagnets such as Mn₃Sn [Nature 527, 212 (2015)] and cubic "hard" antiferromagnets such as Mn₃Pt [Nat. Electron. 1, 172 (2018)]. This work reports the realization of and the detection of spin reversal in a collinear antiferromagnet CuMnAs and thus could be a very good demonstration for many other collinear antiferromagnets. I therefore would like to recommend it for publication in Nature Communications.

I would like to suggest the following two points for revision:

(1) The exchange spring effect used to manipulate the AFM spin axis relies on the spin reversal of the ferromagnetic layer and the devices based on it do not have any advantage beyond conventional ferromagnetic spintronics. But it would be nice that if the authors could add a paragraph to compare the Neel spin-orbit torque [Science 351, 587 (2016) and this work for example] and the electric-field [Nat. Electron. 1, 172 (2018)] approaches in controlling the AFM order parameter, including the disadvantages and advantages for both of them.

(2) The authors could also envision whether it is possible to use the electric-field-induced piezoelectric strain to realize the spin reversal in collinear antiferromagnets as that would be promising for even lower energy-consuming devices.

Reviewer #2 (Remarks to the Author):

In this work, the authors present a new perspective for characterizing the magnetization switching of antiferromagnetic system based on utilizing the broken time and space inversion symmetries. This work largely lowers the threshold for researching of manipulation of the antiferromagnetic devices and digs into the physical nature of the phenomenon. However, before it is accepted, the following questions should be addressed:

1. Since there are multiple electric contacts in the device, the authors should add a brief diagram of the measuring geometry in the photography of the device in Fig.1a.
2. In the last paragraph of page 3, there is the expression 'the device is biased...'. If the word 'biased' can be replaced by stimulated or driven? the 'biased' naturally lead to the idea of a DC current or voltage.
3. In the first paragraph of page 7, the authors declared "the remaining small changes are not observable within the experimental noise on top of the large second harmonic signal but are detectable in the weak first harmonic signal". If this expression refer to Fig. 5a-b, 1ω signal is much larger than 2ω signal in in both absolute value and signal-noise ratio, the authors should give a clear explanation.
4. In the second paragraph of page 8, about the AMR effect, the description of the symmetry of AMR tensor should put at the beginning of this paragraph, the discussion on the feasibility of AMR in Ferromagnetic and Antiferromagnetic system are trivial.
5. Too much space proportion of pictures is occupied by the exhibition of signal stability. Since the

symmetry of the magnetic order lay at the core of this measuring geometry, some visualization of the discussion in Page7-8 is necessary.

6. The meaning of term ξ_{ijk}^{\wedge} at the end of page 8 is not clear.

7. In the supplementary file, part B, the authors should give a clear explanation about the expression for dR_{xx} and dR_{xy} . A consistent deduction is necessary.

8. In the second line of page 7, the word 'surmize' should be corrected .

Reviewer #3 (Remarks to the Author):

This manuscript extends the idea (Ref. 12) of manipulating, via electrical current, the texture of Néel vectors in collinear antiferromagnets (AFM) with broken central symmetry. The underlying mechanism is the staggered spin-orbit (SO) field which can flip the Néel vector direction in monodomain nanoparticles or move the AFM domain wall. One key issue for practical applications here has been reliable detection of AFM states, which differ by only the mutual exchange of sublattice magnetizations. The authors present a solution for this challenge in terms of the tilt induced in the Néel vector via the staggered SO field. Then the measure of a second-order magnetotransport coefficient can reveal the Néel vector direction in the AFM equilibrium state. Experiments presented in the paper clearly demonstrated this approach as well as the long term stability of the second harmonic probing signal. The results are original and expected to have a substantial impact on the research of AFM spintronics.

As a minor point, it would be desirable to estimate the minimal power requirement and the measurement time for realizing the detection of the CuMnAs (or similar material) state. Note also that Refs. 12 and 23 refer to the same article.

Reviewer #1:

Comment

...The detection of spin reversal in antiferromagnets is even more challenging due to their tiny macroscopic moments. Fortunately, the Berry-curvature-induced anomalous Hall effect has enabled the successful detection of the spin reversal via the sign of the Hall resistance in non-collinear antiferromagnets including both hexagonal "soft" antiferromagnets such as Mn₃Sn [Nature 527, 212 (2015)] and cubic "hard" antiferromagnets such as Mn₃Pt [Nat. Electron. 1, 172 (2018)]. This work reports the realization of and the detection of spin reversal in a collinear antiferromagnet CuMnAs and thus could be a very good demonstration for many other collinear antiferromagnets. I therefore would like to recommend it for publication in Nature Communications.

I would like to suggest the following two points for revision:

(1) The exchange spring effect used to manipulate the AFM spin axis relies on the spin reversal of the ferromagnetic layer and the devices based on it do not have any advantage beyond conventional ferromagnetic spintronics. But it would be nice that if the authors could add a paragraph to compare the Neel spin-orbit torque [Science 351, 587 (2016) and this work for example] and the electric-field [Nat. Electron. 1, 172 (2018)] approaches in controlling the AFM order parameter, including the disadvantages and advantages for both of them.

Response

We appreciate that the Reviewer reminded us about the important Ref. [Nat. Electron. 1, 172 (2018)]. In the revised manuscript we included this reference (now Ref. 11) in the modified first paragraph of the Introduction. We also added a more specific comment in the Discussion section which highlights the fundamental distinction between our experiment and the work by Liu et al. In our case, the writing current induces switching between two equally stable antiferromagnetic states with opposite Néel vector directions which are subsequently detected after the writing current has been turned off by measuring the sign flip in the second order magneto-transport. Liu et al., on the other hand, apply a piezovoltage to switch between a zero-voltage stable non-collinear antiferromagnetic state showing a finite anomalous Hall effect, and a transient collinear state stabilized only when the piezovoltage is turned on and showing zero anomalous Hall effect.

We added the following comment in the Discussion:

“Moreover, Liu et al. [Ref. Nat. Electron. 1, 172 (2018)] have shown in M_3Pt that piezoelectric strain can cause a magnetic phase transition from a non-collinear spin arrangement hosting the AHE to a collinear antiferromagnetic phase that does not belong to one of the 31 ferromagnetic symmetry point groups and, correspondingly, shows no AHE.”

(2) The authors could also envision whether it is possible to use the electric-field-induced piezoelectric strain to realize the spin reversal in collinear antiferromagnets as that would be promising for even lower energy-consuming devices.

By symmetry, the piezovoltage does not generate a unidirectional field and we therefore do not envisage direct means of controlling the 180° reversal by this mechanism. This is in contrast to our current-polarity controlled switching in which opposite currents generate opposite (staggered) spin-orbit fields, in analogy to the unidirectional Zeeman-field switching in ferromagnets.

Reviewer #2:

Comment

1. Since there are multiple electric contacts in the device, the authors should add a brief diagram of the measuring geometry in the photograph of the device in Fig. 1a.]

Response

Following this advice, we have marked in the SEM image of the device (Fig. 2a) the measurement set-up for the simultaneous detection of the transverse second-harmonics signal $R_{xy}^{2\omega}$ and the longitudinal first-harmonics signal $R_{xx}^{1\omega}$ and amended the figure caption accordingly.

Comment

2. In the last paragraph of page 3, there is the expression ‘the device is biased...’. If the word ‘biased’ can be replaced by stimulated or driven? the ‘biased’ naturally lead to the idea of a DC current or voltage.

Response

We have changed the original sentence: “In our detection experiments, the device is biased by a low frequency (...) probing current ... with an effective value of” to:

“In our detection experiments, a low frequency(...) probing current ... with an effective value of ... is applied to our device.”

Comment

3. In the first paragraph of page 7, the authors declared "the remaining small changes are not observable within the experimental noise on top of the large second harmonic signal but are detectable in the weak first harmonic signal". If this expression refer to Fig. 5a-b, 1ω signal is much larger than 2ω signal in in both absolute value and signal-noise ratio, the authors should give a clear explanation.

Response

By "large second harmonic signal" we meant that it is not diminished by the randomization of the Néel vector compared to the second harmonic signal of a fully switched state. Similarly, by "weak first harmonic signal" we meant that it is significantly diminished by the randomization of the Néel vector compared to the first harmonic signal of a fully switched state. We agree that this meaning was not clear from our original text. We have changed the text to avoid confusion as follows:

"From the comparison between Figs.~5a and 5b we surmise that a significant randomization within the semicircle is already completed before we initiate the readout measurement ~10 seconds after the pulse was applied. The remaining small changes are then not observable within the experimental noise on top of the second harmonic signal which remains significant compared to its unrelaxed value. On the other hand the changes are detectable in the remaining first harmonic signal which is strongly diminished compared to the first harmonic signal of a fully switched state."

Comment

4. In the second paragraph of page 8, about the AMR effect, the description of the symmetry of AMR tensor should put at the beginning of this paragraph, the discussion on the feasibility of AMR in Ferromagnetic and Antiferromagnetic system are trivial.

Response

Following the Reviewer's suggestion we have revised the paragraphs as follows:

"AMR is a complementary linear-response effect allowing to detect the direction of the order parameter in magnetic films. It has been detected in CuMnAs, as well as in FeRh, MnTe, or Mn₂Au that all host a collinear fully compensated Néel order^{13,16-18,21,22}. However, AMR corresponds to the linear-response magneto-resistance coefficient that is even under time reversal, $\rho_{\text{even}}(\mathbf{O}) = \rho_{\text{even}}(-\mathbf{O})$, i.e., gives the same electrical signal when reversing spins by 180° . This applies equally to any of the 31 ferromagnetic symmetry point groups and also to any of the remaining 91 symmetry point groups of "true" antiferromagnets that do not allow for a net magnetic moment without changing the symmetry of the magnetic crystal.

By measuring the second-order magneto-transport coefficient we can extend the detection of the 180° spin reversal from antiferromagnets within the 31 ferromagnetic point groups to the larger family of antiferromagnetic point groups with broken T-symmetry and no net moment allowed in the point group. In total there is 59 of these broken T-symmetry antiferromagnetic point groups. However, there is an additional symmetry condition required for the presence of the second-order magneto-transport coefficient which is the broken P symmetry in the antiferromagnetic lattice."

Comment

5. Too much space proportion of pictures is occupied by the exhibition of signal stability. Since the symmetry of the magnetic order lay at the core of this measuring geometry, some visualization of the discussion in Page7-8 is necessary.

Response

To better visualize the symmetry discussion we have included Fig. 6 in which we show time reversal and space inversion operations in the antiferromagnetic lattice of CuMnAs.

Comment

6. The meaning of term ξ_{ijk} at the end of page 8 is not clear.

Response

We now define explicitly in the revised text that ξ_{ijk}^{odd} is the second-order transport coefficient (odd under time reversal).

Comment

7. In the supplementary file, part B, the authors should give a clear explanation about the expression for dR_{xx} and dR_{xy} . A consistent deduction is necessary.

Response

We now define in the revised Supplementary information, $\delta R_{ij} = \partial R_{ij} / \partial \varphi \delta \varphi$, with the deflection angle of the Néel vector L from its equilibrium position, $\delta \varphi \sim \pm J_0 \cos(\varphi) \sin(\omega t)$, and φ denoting the angle between the equilibrium Néel vector and the applied current direction.

Comment

8. In the second line of page7, the word 'surmize' should be corrected .

Response

We have corrected the typo.

Reviewer #3:

Comment

As a minor point, it would be desirable to estimate the minimal power requirement and the measurement time for realizing the detection of the CuMnAs (or similar material) state.

Response

We have added a comment in the discussion of Fig. 4d on the power densities corresponding to the range of employed J_{ac} 's and on the detection times used in our experiments. We note here that our devices were designed to make the initial observation of the effect rather than for optimizing the writing and readout energies.

We have added the following remark in the corresponding section:

“We estimate a minimum energy of 1 μ J required for the second-harmonic readout in our experimental setup which was not optimized for minimizing the readout energy. However, increasing the probing current frequency would allow faster second harmonic signal detection and correspondingly lower readout energies.”

Comment

Note also that Refs. 12 and 23 refer to the same article.

Response

We have corrected the list of references.

REVIEWERS' COMMENTS:

Reviewer #1 (Remarks to the Author):

The authors have revised the manuscript based on my suggestion and other reviewers' comments. It is my pleasure to recommend it for publication in Nature Communications. Hope this work could excite more research interest for continuously pushing the antiferromagnetic spintronics forward.

Reviewer #2 (Remarks to the Author):

The authors meet the referee's requirements, so I recommend it to publish on nature communications.

Reviewer #3 (Remarks to the Author):

The authors have successfully addressed the referees' criticisms/comments. I recommend this manuscript be accepted for publication.